# Inhibition of Transglutaminase 2 but Not of MDM2 Has a Significant Therapeutic Effect on Renal Cell Carcinoma

**DOI:** 10.3390/cells9061475

**Published:** 2020-06-16

**Authors:** Joon Hee Kang, Seon-Hyeong Lee, Jae-Seon Lee, Su-Jin Oh, Ji Sun Ha, Hyun-Jung Choi, Soo-Youl Kim

**Affiliations:** Division of Cancer Biology, Research Institute, National Cancer Center, Goyang, Gyeonggi-do 10408, Korea; wnsl2820@gmail.com (J.H.K.); shlee1987@gmail.com (S.-H.L.); ljs891109@gmail.com (J.-S.L.); sujiniii225@gmail.com (S.-J.O.); jsha9595@gmail.com (J.S.H.); labchoihj@gmail.com (H.-J.C.)

**Keywords:** MDM2, p53, transglutaminase 2, nutlin-3a

## Abstract

More than 50% of human cancers harbor *TP53* mutations and increased expression of *Mouse double minute 2 homolog*
*(MDM2)*, which contribute to cancer progression and drug resistance. Renal cell carcinoma (RCC) has an unusually high incidence of wild-type p53, with a mutation rate of less than 4%. MDM2 is master regulator of apoptosis in cancer cells, which is triggered through proteasomal degradation of wild-type p53. Recently, we found that p53 protein levels in RCC are regulated by autophagic degradation. Transglutaminase 2 (TGase 2) was responsible for p53 degradation through this pathway. Knocking down TGase 2 increased p53-mediated apoptosis in RCC. Therefore, we asked whether depleting p53 from RCC cells occurs via MDM2-mediated proteasomal degradation or via TGase 2-mediated autophagic degradation. In vitro gene knockdown experiments revealed that stability of p53 in RCC was inversely related to levels of both MDM2 and TGase 2 protein. Therefore, we examined the therapeutic efficacy of inhibitors of TGase 2 and MDM2 in an in vivo model of RCC. The results showed that inhibiting TGase 2 but not MDM2 had efficient anticancer effects.

## 1. Introduction

Recently, we found that p53 in renal cell carcinoma (RCC) cells can be regulated by both mouse double minute 2 homolog (protein: MDM2; gene: *MDM2*) through proteasomal degradation [1] and by transglutaminase 2 (protein: TGase 2; gene: *TGM2*) through autophagosomal degradation [2]. However, the dominant mechanism that promotes RCC proliferation is unclear.

Human MDM2 (E3 ubiquitin-protein ligase MDM2), encoded by the *MDM2* gene [3], inactivates the p53 tumor suppressor through proteasomal degradation. MDM2 also controls p53 activity by directly binding at ~120 amino acids (a.a.) of the N-terminus of MDM2 with ~30 a.a. of the N-terminus of the transactivation domain of p53 [4,5]. An MDM2 mutant lacking E3 ligase activity also binds wild-type p53 efficiently and inhibits p53 activation [6]. For this reason, interfering with the protein–protein interaction between p53 and MDM2 has been considered as a therapeutic approach for RCC. Indeed, Hoffmann-La Roche has developed a specific small molecule inhibitor, nutlin-3a, which interferes with the MDM2–p53 interaction [7]. About 50% of all human cancers contain an abnormal *TP53* gene [8]. By contrast, almost all RCC cells harbor wild-type p53; indeed, only 4% of RCC harbor mutations in *TP53* [2]. Therefore, MDM2-mediated proteasomal degradation must be a major regulator of wild-type p53 in RCC. However, although nutlin-3a activates p53 and inhibits cell proliferation in nontumorigenic NIH-3T3 cells, it fails to kill the cells [9]. Nutlin-3a alone did not induce cell death in a xenograft model of human breast cancer cells [10]; however, it acted synergistically with carboplatin to exert anticancer effects [10]. To overcome this issue, several isotypes of nutlin-3a were developed for clinical trials: these include RG7112 from Hoffmann-La Roche [11], AMG-232 from Amgen [12], NVP-CGM097 from Novartis [13], and MK-8242 from Merck [14].

Recently, we found that TGase 2 (E.C. 2.1.2.13) plays a major role in regulating p53 in RCC [2,15,16,17,18]. TGase 2 is a calcium enzyme that cross-links enzyme protein-bound glutamine and lysine to form covalent ε-(γ-glutamyl)lysine [19,20,21,22]. We found that TGase 2 acts like a chaperone to transfer binding proteins to a specific location via a triple complex [2]. A series of reports shows that *TGM2*knockdown using siRNA induces cell death in RCC cell lines but not in normal immortalized cells HEK293 [2,15,16,17] or other cancer cell lines [23,24]. This implies that RCC specifically employs TGase 2 for survival. In addition, TGase 2 acts as a chaperone that carries proteins without cross-linking them [25]. TGase 2 forms a ternary complex wherein the N-terminus of p53 binds to the N-terminus of TGase 2 while the C-terminus of TGase 2 binds the N-terminus of p62 [2]; p53 is transported automatically to autophagosomes that contain microtubule-associated protein 1A/1B-light chain 3 (LC3), a p62 receptor which triggers autophagic degradation following complex formation [2]. Knocking down *TGM2* or inhibiting TGase 2p53 binding in RCC stabilizes p53, thereby inducing p53-mediated cell death. We demonstrated that blocking the interaction between TGase 2 and p53 with streptonigrin stabilizes p53 to induce apoptosis in RCC cell lines [15]. Another study showed that wild-type p53 in RCC cells is functional and transcriptionally active and that it responds normally to DNA damage induction by UV radiation [26].

The aim of the present study was 2-fold: first, we asked whether destabilization of p53 in vitro is dependent on MDM2-mediated proteasomal degradation or TGase 2-mediated autophagic degradation; second, we asked whether inhibiting MDM2 or TGase 2 in an in vivo RCC model has anticancer effects.

## 2. Materials and Methods

### 2.1. Antibodies and Reagents

The following antibodies were used: TGase 2 (Cat. #MS-300-P0, Thermo Scientific, Waltham, MA, USA, 1:1000 and Cat. #SAB4200073, Sigma Aldrich, St. Louis, MO, USA, 1:2000 for immunohistochemistry), β-actin (Cat. #sc-47778, Santa Cruz Biotechnology, Dallas, TX, USA, 1:1000), p53 (Cat. #sc-126, 1:1000 and Cat. #sc-6243, 1:1000, Santa Cruz Biotechnology), and MDM2 (Cat. #sc-813, Santa Cruz Biotechnology, 1:1000). Antibodies specific for Ki67 (Cat. #ab15580, Abcam, Cambridge, UK, 1:3000), streptonigrin (Cat. #S1014), and nutlin-3a (Cat. #SML0580) were purchased from Sigma-Aldrich (St. Louis, MO, USA). The INTERFERin^®^ (Cat. #409-50) transfection reagent was from Polyplus-trasnfection Co. (Illkirch-Graffenstaden, FRA). A small interfering RNA (siRNA) duplex targeting human *TG*M2 or *MDM2* was purchased from GenePharma (Shanghai, CN).

### 2.2. Cell Culture

RCC cell lines ACHN and CAKI-1 were obtained from the National Cancer Institute (Material Transfer Agreement number: 2702-09). Cells were cultured at 37 °C in complete RPMI 1640 medium (Hyclone, UT, USA) containing 10% fetal bovine serum (Hyclone, UT, USA) in an atmosphere of 5% CO_2_ (100% humidity).

### 2.3. Western Blot Analysis

For western blot analysis, cells were lysed using RIPA buffer and protein assays were carried out to normalize the protein content (Bicinchoninic acid protein assay kit; Pierce, Rockford, IL, USA). Then, 10 μg total protein was separated in SDS-polyacrylamide gels and transferred to polyvinylidene fluoride membranes. The membranes were incubated for 1 h with 5% bovine serum albumin in TBST (Tris-buffered saline/tween, 50 mM Tris-Cl, pH 7.5. 150 mM NaCl.0.1% Tween 20) and then incubated (1 h 30 min) at room temperature with the indicated antibodies. Primary antibodies specific for TGase2, p53, MDM2, and β-actin were used at a dilution of 1:1000. After three washes with TBST, membranes were incubated for 1 h at room temperature with an horseradish peroxidase-conjugated secondary antibody. Membranes were washed five times with TBST, and chemiluminescence was detected using Westsave™ (Abfrontier, KOR). Gels were imaged using FUSION-Solo.4.WL (Vilber Lourmat, FRA).

### 2.4. Real-Time Apoptosis Assay

ACHN and CAKI-1 cells were seeded in white 96-well culture plates (10,000 cells/well; 50 μL/well) and incubated overnight until they adhered to the plastic. The next day, the test compounds were prepared as 4× solutions in the medium and the assay reagents were prepared according to the manufacturer’s protocol (Promega, Cat#: JA1011, Madison, WI, USA). The compounds were added (25 mL/well) to the test wells, followed by the assay reagents (25 mL/well). The plate was incubated immediately at 37 °C in a plate reader fitted with a humidity chamber. Membrane integrity was measured as a fluorescence signal (ex/em: 485 ± 10/525 ± 10), and phosphatidylserine translocation was measured as a luminescence signal (counts/s; integration time, 1000 ms) generated by the annexin V-dependent assembly of two fragments of luciferase. Measurements were performed after 24 h.

### 2.5. Preclinical Xenograft Tumor Models

Six-week-old, female, specific pathogen-free Bagg and Albino (BALB)/c nude mice (*n* = 12) were purchased from Central Lab (Animal Inc., Seoul, Korea) and injected subcutaneously with ACHN cells (5.0 × 10^6^ cells/head). When tumors reached an appropriate volume of 150–200 mm^3^, mice were randomized into three groups (*n* = 4–5/group) according to tumor volume and body weight. The control group was treated with vehicle only (0.04% dimethyl sulfoxide (DMSO) in Phosphate-buffered saline); the streptonigrin-treated group received 0.1 mg/kg of the compound; and the nutlin-3a-treated group received 20 mg/kg of the compound. Vehicle and streptonigrin were administered orally once per day (5 days/week). Nutlin-3a was administered once per day (5 days/week) by intraperitoneal injection. The size of the primary tumors was measured every 3–4 days using calipers. Tumor volume was calculated using the formula V = (A × B2)/2, where V is the volume (mm^3^), A is the long diameter, and B is the short diameter (mm). Mice were euthanized using 7.5% CO_2_, and tumors were harvested for immunohistochemical analysis. The study was reviewed and approved by the Institutional Animal Care and Use Committee (IACUC) of the National Cancer Center Research Institute (NCCRI) in the Republic of Korea. The NCCRI is an Association for Assessment and Accreditation of Laboratory Animal Care International (AAALAC International)-accredited facility and abides by the Institute of Laboratory Animal Resources (ILAR) guidelines (Institutional Review Board number: NCC-19-509, approval date. 09.12.19).

### 2.6. Automated Immunohistochemistry

Immunohistochemistry assays were performed using a VENTANA Discovery XT automated staining instrument according to the manufacturer’s instructions. For tumor staining, slides were prepared from the xenograft and de-paraffinized for 30 min at 75 °C using EZprep solution (Ventana Medical Systems, AZ, USA). Epitope retrieval was accomplished on the automated stainer by exposure to CC2 solution (Ventana Medical Systems, AZ, USA) for 64 min at 95 °C. Antibodies were first titered over a range of concentrations to identify the optimal ratio for specific-to-background staining. Once the titers were set, the antibodies in a diluent were transferred to user-fillable dispensers for use on the automated stainer. Antibodies specific for Ki67, p53, and TGase 2 were used. Slides were developed using the Optiview diaminobenzidine (DAB) detection kit (Ventana Medical Systems, AZ, USA). Briefly, the steps were as follows: inhibitor for 8 min, linker for 8 min, multimer for 12 min, DAB/peroxide for 8 min, and copper for 4 min. Slides were then counterstained for 8 min with hematoxylin II (Ventana Medical Systems, AZ, USA). The titers of each antibody were measured using positive and negative control tissues according to the manufacturer’s instructions. Representative images from each tumor were collected under a microscope (Scanscope XT, Leica, DE) fitted with a 20× objective lens.

### 2.7. Statistical Analysis

Statistical analysis was performed using Student’s t-test. Tumor growth in the xenograft mouse model was analyzed statistically by two-way analysis of variance (ANOVA) using Microsoft Excel. A *p*-value less than 0.05 (typically ≤0.05) is statistically significant.

## 3. Results

### 3.1. TGase 2 and MDM2 Suppress p53 in RCC

Previously, we reported that p53 is regulated competitively by TGase 2 and MDM2 in RCC under starvation conditions. To evaluate the contribution of TGase 2 and MDM2 to p53 stabilization, we introduced small interfering (si)RNAs specific for *TGM2* and *MDM2* into ACHN and CAKI-1 for 24 h (Figure 1A, Appendix A). The amount of p53 increased about 7.4-fold when TGase 2 was silenced using 40 nM siRNA targeting *TGM2* and by about 3.1-fold when *MDM2* was silenced using 40 nM siRNA targeting *MDM2* in ACHN cells (Figure 1B). A similar pattern was observed in CAKI-1 cells with 5.1-fold and 1.5-fold increases, respectively (Figure 1B, Appendix A). This result suggests that both TGase 2 and MDM2 regulate p53 stability in RCC, although TGase 2 appears to be the stronger regulator under normal conditions (37 °C, 5% CO_2_, and 21% O_2_). To evaluate the contribution of TGase 2 and MDM2 to apoptosis through p53 stabilization, apoptosis was measured in a real-time apoptosis assay (Figure 1C, Appendix A). The results showed that knockdown of *TGM2* and *MDM2* increased the number of annexin V-positive ACHN and CAKI-1 cells in a dose-dependent manner (Figure 1C).

### 3.2. p53 in RCC is Stabilized by Treatment with Streptonigrin and Nutlin-3a

To investigate the effects of TGase 2 and MDM2 on p53 protein levels in RCC, we targeted them using streptonigrin (which inhibits TGase 2-p53 binding) [15] and nutlin-3a (which inhibits MDM2-p53 binding) [7] (Figure 2A, Appendix A). The level of p53 protein in ACHN cells increased by 14-fold after exposure to 100 nM streptonigrin. After nutlin-3a treatment (10 μM), p53 levels increased by 8.7-fold (Figure 2B, Appendix A). In CAKI-1 cells, p53 protein levels increased by 3.9- and 3.0-fold after treatment with streptonigrin (100 nM) or nutlin-3a (10 μM), respectively. Next, we examined apoptosis in vitro (Figure 2C, Appendix A). Nutlin-3a (at 10 μM) triggered a ~2-fold increase in apoptosis, while streptonigrin (10 nM) also increased it by ~ 2-fold (Figure 2C). The results showed that nutlin-3a had the same effect as streptonigrin but at a 1000-fold higher dose (Figure 2C).

### 3.3. TGase 2 mRNA Levels Increase in RCC

To analyze the association between RCC and expression of TGase 2 and MDM2, we analyzed TGase 2 and MDM2 transcripts using the U.S. National Cancer Institute database. The microarray data showed that *TGM2* mRNA levels increased in all RCC cell lines (Figure 3A, top panel). Previously, we reported that the TGase 2 autophagosome degrades p53 by binding to SQSTM1/p62. Data from the database shows that the level of transcripts encoding the autophagy marker *SQSTM1/p62* increased in most RCC cell lines (Figure 3A, middle). The expression of *MDM2* mRNA was not consistent between RCC cell lines (Figure 3A, bottom). We examined the association between overall survival of patients with RCC and expression of TGase 2 or MDM2 (Figure 3B). We found that cancer patients with high expression of TGase 2 had poorer survival than patients with low expression. This difference was not as marked for MDM2 (Figure 3B).

### 3.4. Inhibiting TGase 2 Suppresses the Growth of RCC in a Xenograft Model

To analyze the anticancer effects of streptonigrin or nutlin-3a, we tested whether they exert therapeutic effects in a xenograft model of human RCC. A single treatment with streptonigrin inhibited ACHN cell growth, whereas nutlin-3a monotherapy had no effect (Figure 4A, Appendix A, Appendix A). Immunohistochemical staining of TGase 2, p53, and Ki67 was performed in tumors collected at the end of the animal experiments (Figure 4B, Appendix A, Appendix A). The results showed a significant increase in p53 expression in tumors from streptonigrin-treated mice compared with the expression in control and nutlin-3a-treated mice (Figure 4B). Ki67 expression showed a significant decrease after streptonigrin treatment but no change after nutlin-3a treatment (Figure 4B). The level of TGase 2 expression was not changed by streptonigrin treatment but increased after nutlin-3a treatment (Figure 4B). As shown in Figure 1 and Figure 2, early inhibition of MDM2 increased p53 expression. However, there was no difference in the xenograft model. To understand these phenomena, ACHN and CAKI-1 cells were treated with nutlin-3a (10 mM) in a time-dependent manner. The expression of MDM2 was unchanged for 4 days after nutlin-3a treatment (Figure 5A, Appendix A). However, the expression of TGase 2 was induced about 1.5-fold after nutlin-3a treatment for 4 days. The expression of p53 was increased up to 3 days post-nutlin-3a treatment but decreased to about 20% at day 4 (Figure 5A). A simple graphic showing the mechanism underlying p53 regulation by TGase 2 and MDM2 in renal cancer is shown in Figure 5B,C. In this study, we did not observe increased expression of p53, MDM2, or p21 in mice treated with nutlin-3a, although previous reports suggest an increase (Figure 5B,C) [9,27].

## 4. Discussion

In vitro knockdown of *MDM2* or *TGM2* using siRNA showed that both induced an ~2-fold increase in induction of apoptosis within 24 h under normoxic culture conditions. However, inhibiting TGase 2 using streptonigrin required a dose that was 1000-fold less than that required of nutlin-3a to inhibit MDM2. Furthermore, 20 mg/kg of nutlin-3a did not inhibit tumor growth, whereas 0.1 mg/kg of streptonigrin inhibited growth by almost 70%. Long-term treatment of RCC cells with nutlin-3a for 4 days increased TGase 2 expression concomitantly with a decrease in p53 levels (Figure 5A). Therefore, nutlin-3a treatment may induce TGase 2 expression, which may in turn regulate p53 (Figure 5B,C). This suggests that TGase 2 may be responsible for the reverse effects of nutlin-3a, although it is unclear how nutlin-3a induces TGase 2 in RCC. Therefore, nutlin-3a combined with a TGase 2 inhibitor may be an effective anticancer therapy.

The p53 protein induces *MDM2* at the transcriptional level, and the MDM2 protein regulates p53 protein at the protein level [28]. This suggests that an increased expression of p53 activity by nutlin-3a may induce expression of *MDM2*, which feeds back to regulate the protein level of p53. Therefore, long-term treatment with nutlin-3a altered the expression of p21 or p53 proteins. This suggests that inhibiting MDM2 using nutlin-3a may induce MDM2 activity by liberating it from p53 [29]. MDM2 promotes genomic instability and cell proliferation by binding to tumor suppressors such as p21, Rb, hnRNP K, and E-cadherin (reviewed in References [29,30]). MDM2 degrades cell-cycle inhibitors such as p21 and hnRNP K, which promote cell proliferation [31]. Degradation of p21 by MDM2 also induces genomic aberrations due to the role of p21 in maintaining genomic stability [27]. MDM2 binds and degrades Rb directly, thereby inhibiting the Rb-E2F1 interaction and promoting transcription of E2F1 to drive cancer cell proliferation [32]. MDM2 degrades E-cadherin to drive epithelial–mesenchymal transition and tumorigenesis [30]. Another explanation may be that activation of p53 by nutlin-3a activates p53 target genes such as *MDM2* [9]. Therefore, continuous activation of p53 by inhibiting MDM2 may initiate a feedback to stop by *MDM2* induction. However, we did not observe any changes in the expressions of MDM2 or p21 or any apoptosis in the mouse group treated with nutlin-3a.

By contrast, recent evidence suggests that TGase 2 may be a good therapeutic target for RCC [33]. In RCC, TGase 2 is negatively regulated by the von Hippel–Lindau tumor suppressor protein (pVHL) and positively regulated by HIF-1α. pVHL and HIF-1α are the most influential regulator and promoter of RCC, respectively. Therefore, mutation of *VHL* and hypoxic conditions induce continuous expression of TGase 2, which then depletes p53 via autophagic degradation. This is why inhibiting TGase 2 may be a beneficial therapeutic approach.

The most important limitation of therapies for RCC is the emergence of a recurrence phenotype that endows drug resistance [34]. The primary treatments for RCC are sorafenib and sunitinib [35], which inhibit tyrosine kinases [36]. Autophagy is a protective mechanism that promotes a resistance phenotype in cancer cells during chemotherapy [37]. Inhibiting autophagy sensitizes resistant cancer cells to chemotherapeutic drugs [37], which suggests that inhibiting autophagy may overcome drug resistance. Although everolimus (an mTOR inhibitor) is considered a second-line treatment for RCC after sunitinib treatment failure, increasing drug resistance is a problem [38]. Since inhibiting mTOR induces autophagy, activating autophagy may be a key mechanism underlying resistance to everolimus [39]. Everolimus combined with chloroquine (an autophagy inhibitor) can markedly inhibit autophagic flux and promote apoptosis, suggesting that combined use of targeted therapeutics can improve the therapeutic effects against RCC [40]. Interestingly, the increase in TGase 2 expression in various cancers is associated with increased autophagy due to increased LC3 levels [18]. Furthermore, TGase 2-mediated suppression of p53 induces drug resistance by triggering autophagy via induction of LC3 [18]. Therefore, inhibiting TGase 2 may prevent drug resistance during cancer therapy. A previous report identified rapamycin-enhanced changes in transcription in TSC-deficient cancer cells. *TGM2* mRNA expression increased significantly; indeed, it was one of the 20 transcripts showing the greatest increases in expression among 39,000 probed genes [41]. Interestingly, TGase 2 is also induced by rapamycin in a time-dependent manner [41]. TGase 2 promotes autophagosome maturation. Knocking out *TGM2* impairs maturation of the autophagosome [42]. This suggests that TGase 2-mediated protein pathways are required for maturation of autophagosomes, accompanied by accumulation of damaged proteins. We have shown that TGase 2 transports p53 from the cytosol to the autophagosome through a ternary complex (p53-TGase 2-p62) [2]. This process is not required for ubiquitination of p53 because TGase 2 binds two proteins, p53 and p62, directly [2]. This process is probably preferred by RCC because it consumes zero calories; ubiquitination requires consumption of ATP. Stress induced by DNA damage stabilizes p53 through S^15^ phosphorylation. TGase 2 cannot bind to p53 when S^15^ is phosphorylated [2]. Therefore, increased p53 stability due to TGase 2 inhibition synergistically increases the anticancer effects against RCC treated with DNA-damaging anticancer drugs such as doxorubicin [2].

## 5. Conclusions

TGase 2 knockdown or inhibition induces cell death in RCC cells but not in normal cells [15,16,43]. Here, we show that targeting TGase 2 induces a better therapeutic response than targeting MDM2 because MDM2 inhibitor treatment increases the level of TGase 2. Therefore, receptor tyrosine kinase combined therapy with inhibitor of TGase 2 [15,44,45] may be an effective approach to treating RCC.

## Figures and Tables

**Figure 1 cells-09-01475-f001:**
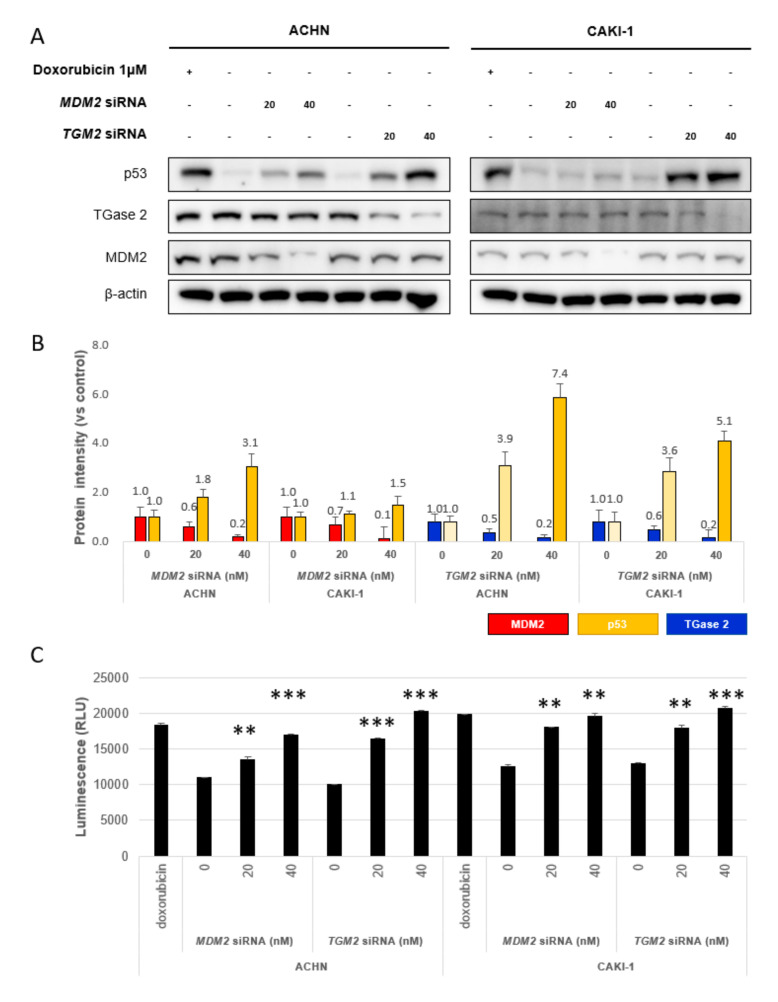
Transglutaminase 2 (TGase 2) and Mouse double minute 2 homolog (MDM2) destabilize p53 in renal cell carcinoma (RCC) under normoxic conditions. (**A**) ACHN (left) and CAKI-1 (right) cells were transfected for 24 h with siRNA targeting *TGM2* or *MDM2*. Doxorubicin (1 μM) was used as a positive control. (**B**) Image J analysis of western blots in A. (**C**) Apoptosis assay to assess *siTGM2* or *siMDM2* silencing, as monitored by RealTime-Glo™ Annexin V under the same conditions as in A. Cumulative data from three independent experiments are shown (mean ± SD; *n* = 3). ***p* < 0.01 and ****p* < 0.001.

**Figure 2 cells-09-01475-f002:**
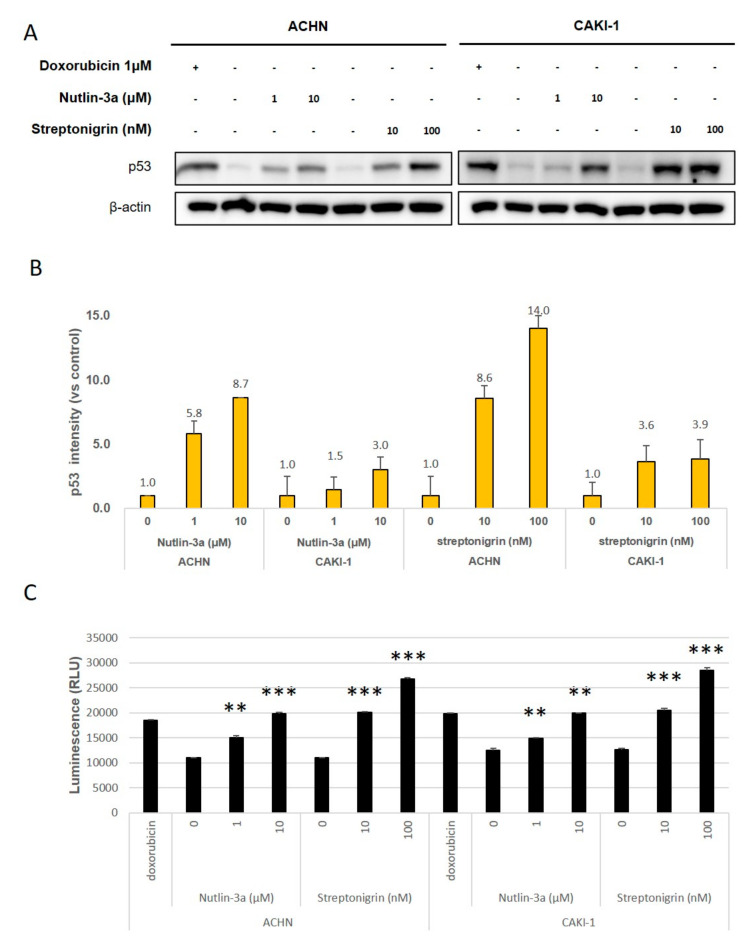
Streptonigrin and nutlin-3a increase p53 protein levels in a dose-dependent manner. (**A**) ACHN (left) and CAKI-1 (right) cells were treated with streptonigrin (which inhibits TGase 2-p53 binding) or nutlin-3a (which inhibits MDM2-p53 binding) for 24 h. Doxorubicin (1 μM) was used as a positive control. (**B**) Image J analysis of the western blots in A. (**C**) The apoptosis assay was performed in the presence/absence of streptonigrin or nutlin-3a and was monitored by RealTime-Glo™ annexin V under the conditions outlined in A. Cumulative data from three independent experiments are shown (mean ± SD; *n* = 3). ***p* < 0.01 and ****p* < 0.001.

**Figure 3 cells-09-01475-f003:**
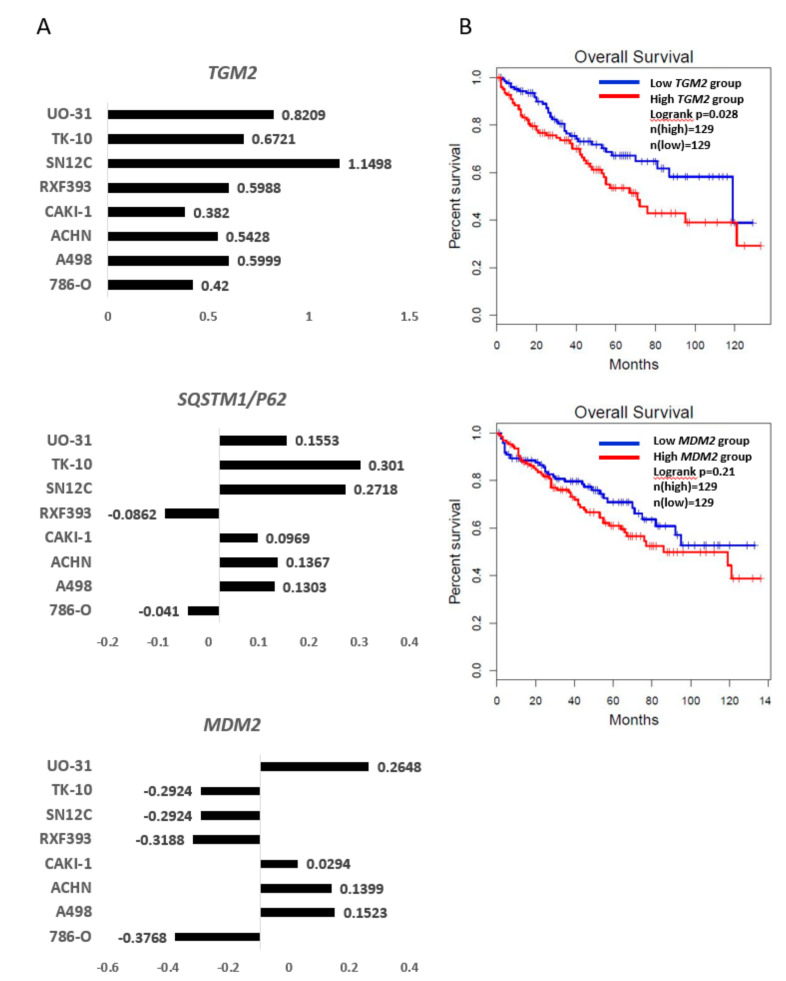
Expression of *TGM2*, *MDM2*, and *SQSTM1/p62* in RCC: (**A**) Microarray data from the U.S. National Cancer Institute (http://dtp.nci.nih.gov/mtweb/targetdata). *TGM2* (experiment ID: 9993; https://dtp.cancer.gov/mtweb/targetinfo?moltid=GC19395&moltnbr=9993) expression is consistently higher in RCC cell lines, whereas that of *MDM2* (experiment ID: 3080; https://dtp.cancer.gov/mtweb/targetinfo?moltid=GC12482&moltnbr=3080) expression is not. Expression of *SQSTM1/P62* (experiment ID: 9233; https://dtp.cancer.gov/mtweb/targetinfo?moltid=GC18635&moltnbr=9233) is higher in most RCC cell lines. The graph is on a log scale (mRNA levels in cell line/mRNA levels in a reference pool). Reference probes were made by pooling equal amounts of mRNA from the HL-60, K562, NCI-H226, COLO205, SNB-19, LOX-IMVI, OVCAR-3, OVCAR-4, CAKI-1, PC-3, MCF7, and Hs578T cell lines. The average mRNA levels of *TGM2, SQSTM1/P62*, and *MDM2* are 0.05, −0.05, and −0.09, respectively. (**B**) Kaplan–Meier survival curves based on expression of *TGM2* or *MDM2*: Patients with high expressions of *TGM 2* show shorter overall survival than those with low expression (*p* = 0.028). The overall survival of *MDM2* is not significant (*p* = 0.21).

**Figure 4 cells-09-01475-f004:**
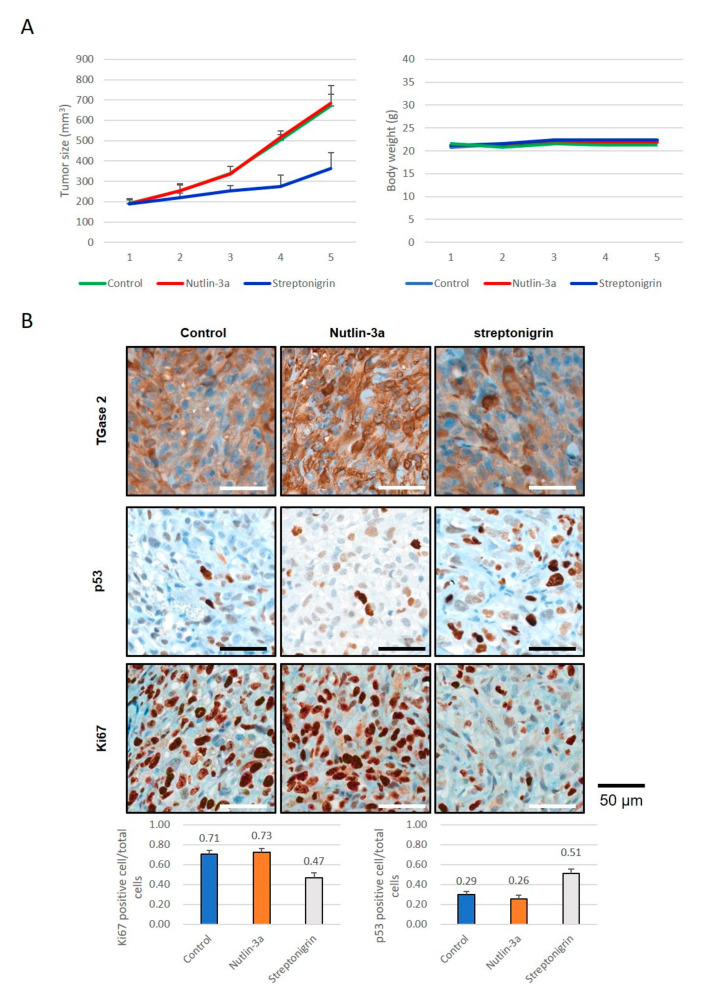
The anticancer effects of inhibition of TGase 2 and MDM2 in a human RCC xenograft model: (**A**) ACHN cells were injected subcutaneously into one flank of BALB/c nude mice (*n* = 5). Mice received streptonigrin or nutlin-3a when tumors reached a volume of 200 mm^3^. The average tumor volume and body weight is presented as the mean ± SE. (**B**) ACHN cells harvested from BALB/c nude mice were analyzed by immunohistochemical staining for Ki67, TGase 2, and p53. The bar graph represents the percentage of cells positive for Ki67 or p53. Cumulative data from three independent experiments are shown (mean ± SD; *n* = 3). ***p* < 0.01 and ****p* < 0.001.

**Figure 5 cells-09-01475-f005:**
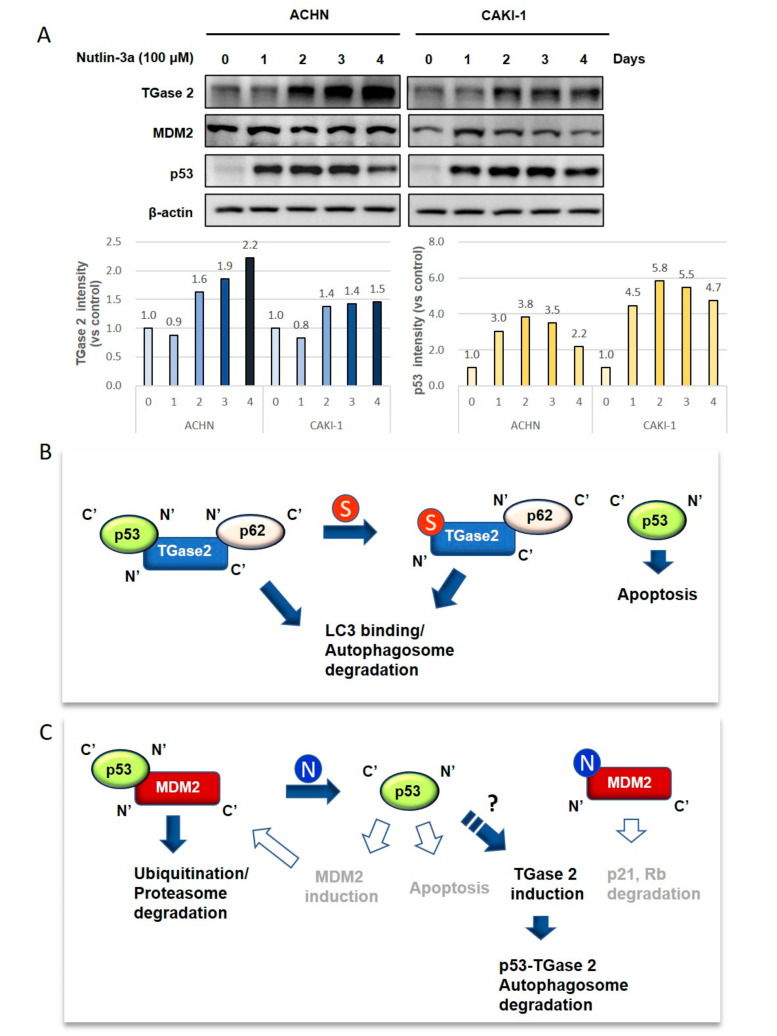
The proposed model explaining p53 regulation by TGase 2 and MDM2 in RCC: (**A**) TGase 2 expression was induced after MDM2 inhibition. ACHN and CAKI-1 cell were treated with nutlin-3a (10 μM) in a time-dependent manner. (**B**) TGase 2 forms a triple complex with p53 and p62, which then moves to the autophagosome via binding of p62 to LC3. [15]. Next, p53 is degraded by the autophagic process. Streptonigrin inhibits TGase 2-p53 binding, which stabilizes p53 and triggers p53-mediated cell death. (**C**) P53-MDM2 binding causes proteasomal degradation of p53 via ubiquitination. Nutlin-3a inhibits the MDM2–p53 interaction, thereby inducing p53-mediated apoptosis. However, activated p53 may induce *MDM2* transcription, which regulates p53 activity. This creates a feedback loop between p53 and MDM2 [28]. Therefore, MDM2 inhibition may have a temporal effect on p53 activation. The released MDM2 also triggers degradation of tumor suppressors such as p21 and Rb, which can lead to uncontrolled cell growth. In this study, we observed only increased TGase 2 expression after nultlin-3a treatment, which is likely induced by p53 activity (bold). S, streptonigrin; N, nutlin-3a; closed arrows and bold font denote observations from this study; and open arrows and gray font denote information obtained from references but failed to observe changes (data not shown). LC3, microtubule-associated proteins 1A/1B light chain 3.

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
