# Peer review of "Inhibition of Transglutaminase 2 but Not of MDM2 Has a Significant Therapeutic Effect on Renal Cell Carcinoma"

_cells, 2020, doi:10.3390/cells9061475_

Round 1

Reviewer 1 Report

In the paper “ Inhibition of transglutaminase 2 but not of MDM2  has a significant therapeutic effect on renal cell 3 carcinoma”. The authors discussed about TP53 mutations and increased expression of  Mouse double minute 2 homolog (MDM2), which contribute to cancer progression.

In a recent work, the authors show their interest on the mechanisms which are able to regulate p53 by autophagic degradation. Transglutaminase 2 (TGase 2) participate to p53 degradation regulating this pathway.

This study is noteworthy to stimulate discussion and debate about the regulation of autophagy in cancer cells. However some points of criticism persist.

The authors reported both Nutlin and Streptonigrin effects (for Streptonigrin the inhibition of TGase 2-p53 binding for Nutlin the inhibition of MDM2-p53 binding ) as phenomena developing in a dose-dependent manner. The effects only of two concentrations have been shown, other results could be added.

Author Response

Reviewer 1

The authors reported both Nutlin and Streptonigrin effects (for Streptonigrin the inhibition of TGase 2-p53 binding for Nutlin the inhibition of MDM2-p53 binding) as phenomena developing in a dose-dependent manner. The effects only of two concentrations have been shown, other results could be added.

Ans) We did not test other concentrations of nutlin-3a and streptonigrin because purpose of this experiments was seeing dose-response instead of analysis of IC50. Beyond 10 mM of nutlin-3a and 100 nM of streptonigrin, RCC cells were all dead. Therefore, we could not raise concentrations.

Reviewer 2 Report

The manuscript by Joon Hee Kang and co-authors addresses the role of TGase 2 and MDM2 in p53 - response pathway and potential therapeutic effect of TGase 2 inhibition in RCC. The link between TGase 2 and p53 is nicely illustrated by in vitro and in vivo experiments. it would be good to show lack of TGase 2 effect on p53 in non RCC cell lines, at least in knockdown experiments, since it seems that TGase 2 engagement in cell survival is cell specific. The MDM2 downregulation effect on p53 is less pronounced than TGase 2 knockdown in RCC model. It is known that MDM2 is one of a major regulator of p53 so please explain the results from present study (discussion section).

Technical issues
In Methods Section:

Lack of cell transfection description. No information how many times the knockdown experiments were repeated.

Lack of description of analysis of data from U.S. National Cancer Institute database.

Figure 1A and 2A: no statistic data (SD, P value).

Figure 3: too small fonts.

Figure 5B and 5C: Proposed model is a bit confusing, no clear which data are from present study.

Author Response

Reviewer 2

it would be good to show lack of TGase 2 effect on p53 in non RCC cell lines, at least in knockdown experiments, since it seems that TGase 2 engagement in cell survival is cell specific. The MDM2 downregulation effect on p53 is less pronounced than TGase 2 knockdown in RCC model. It is known that MDM2 is one of a major regulator of p53 so please explain the results from present study (discussion section).

Ans) We have mentioned that this is RCC specific in the introduction p2.

“A series of reports shows that TGase 2 knockdown using siRNA induces cell death in RCC cell lines, but not in normal immortalized cells HEK293 [2,15-18] or other cancer cell lines [23,24].”

We have mentioned that TGase 2 may be useful for regulating p53 in cancer because TGase 2 is induced by MDM2 inhibition. It has mentioned in the discussion, p11.

“Long term treatment of RCC cells with nutlin-3a for 4 days increased TGase 2 expression concomitantly with a decrease in p53 levels (Figure 5A). Therefore, nutlin-3a treatment may induce TGase 2 expression, which may in turn regulate p53 (Figure 5B and C). This suggests that TGase 2 may be responsible for the reverse effects of nutlin-3a, although it is unclear how nutlin-3a induces TGase 2 in RCC. Therefore, nutlin-3a combined with a TGase 2 inhibitor may be an effective anti-cancer therapy.”

We have added a sentence in the conclusion, p14.

“TGase 2 knockdown or inhibition induces cell death in RCC cells but not in normal cells [15,16,44]. Here, we show that targeting TGase 2 induces a better therapeutic response than targeting MDM2 because MDM2 inhibitor treatment increases the level of TGase 2. Therefore, RTK combined therapy with inhibitor of p53-TGase 2 binding [15,45,46] may be an effective approach to treating RCC.”

Technical issues
In Methods Section:

Lack of cell transfection description. No information how many times the knockdown experiments were repeated. Figure 1A and 2A: no statistic data (SD, P value).

Ans) We have inserted the number of experiments, SD and p values in legends of Figure 1 and 2. Standard bar is marked on the new Figures 1 and 2.

Lack of description of analysis of data from U.S. National Cancer Institute database.

Ans) we have changed the Figure 3 and legend as

“Figure 3. Expression of TGase 2, MDM2, and SQSTM1/p62 in RCC. (A) Microarray data from the U.S. National Cancer Institute (http://dtp.nci.nih.gov/mtweb/targetdata). TGM2 (Experiment Id: 9993; https://dtp.cancer.gov/mtweb/targetinfo?moltid=GC19395&moltnbr=9993) expression is consistently higher in RCC cell lines, whereas that of MDM2 (Experiment Id: 3080; https://dtp.cancer.gov/mtweb/targetinfo?moltid=GC12482&moltnbr=3080) expression is not. Expression of SQSTM1/P62 (Experiment Id: 9233; https://dtp.cancer.gov/mtweb/targetinfo?moltid=GC18635&moltnbr=9233) is higher in most RCC cell lines. The graph is on a log scale (mRNA levels in cell line/mRNA levels in a reference pool). Reference probes were made by pooling equal amounts of mRNA from HL-60, K562, NCI-H226, COLO205, SNB-19, LOX-IMVI, OVCAR-3, OVCAR-4, CAKI-1, PC-3, MCF7 and Hs578T cell lines. TGM2, SQSTM1/P62 and MDM2 average mRNA levels are 0.05, -0.05, -0.09, respectively. (B) Kaplan–Meier survival curves based on expression of TGM2 or MDM2. Patients with high expression of TGM 2 show shorter overall survival than those with low expression (p = 0.028). The overall survival of MDM2 is not significant (p=0.21).”

Figure 3: too small fonts:

Ans) Figure 3 is made up with new picture with bigger font.

Figure 5B and 5C: Proposed model is a bit confusing, no clear which data are from present study.

Ans) It has been introduced in the legend, p13, as follows.

“P53-MDM2 binding causes proteasomal degradation of p53 via ubiquitination. Nutlin-3a inhibits the MDM2-p53 interaction, thereby inducing p53-mediated apoptosis. However, activated p53 may induce MDM2 transcription, which regulates p53 activity. This creates a feedback loop between p53 and MDM2 [28]. Therefore, MDM2 inhibition may have a temporal effect on p53 activation. The released MDM2 also triggers degradation of tumor suppressors such as p21 and Rb, which can lead to uncontrolled cell growth. In this study, we observed only increased TGase 2 expression after nultlin-3a treatment, which is likely induced by p53 activity (bold). S, streptonigrin; N, nutlin-3a; closed arrows and bold font denote observations from this study; open arrows and gray font denote information obtained from references but failed to observe changes (data not shown).”
